# MULTILINGUAL ALIGNMENT OF CONTEXTUAL WORD REPRESENTATIONS

**Steven Cao, Nikita Kitaev & Dan Klein**
Computer Science Division
University of California, Berkeley
{stevencao,kitaev,klein}@berkeley.edu

## ABSTRACT

We propose procedures for evaluating and strengthening contextual embedding alignment and show that they are useful in analyzing and improving multilingual BERT. In particular, after our proposed alignment procedure, BERT exhibits significantly improved zero-shot performance on XNLI compared to the base model, remarkably matching pseudo-fully-supervised translate-train models for Bulgarian and Greek. Further, to measure the degree of alignment, we introduce a contextual version of word retrieval and show that it correlates well with downstream zero-shot transfer. Using this word retrieval task, we also analyze BERT and find that it exhibits systematic deficiencies, e.g. worse alignment for open-class parts-of-speech and word pairs written in different scripts, that are corrected by the alignment procedure. These results support contextual alignment as a useful concept for understanding large multilingual pre-trained models.

## 1 INTRODUCTION

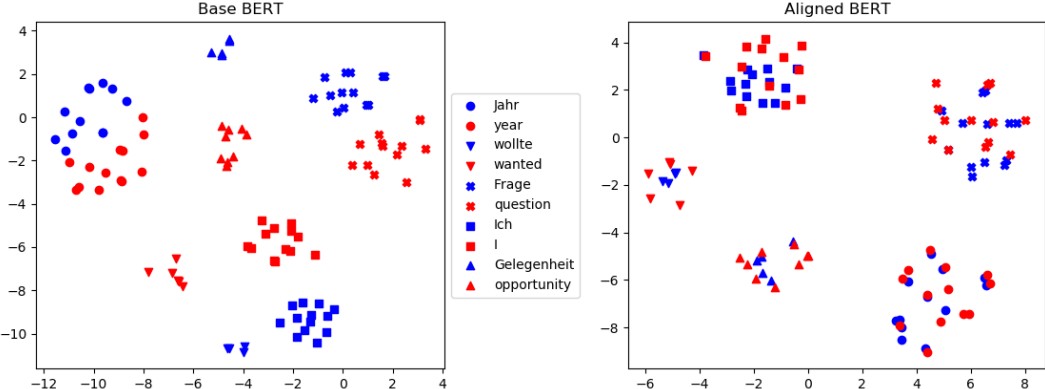

Figure 1: t-SNE (Maaten & Hinton, 2008) visualization of the embedding space of multilingual BERT for English-German word pairs (left: pre-alignment, right: post-alignment). Each point is a different instance of the word in the Europarl corpus. This figure suggests that BERT begins already somewhat aligned out-of-the-box but becomes much more aligned after our proposed procedure.

Embedding alignment was originally studied for word vectors with the goal of enabling cross-lingual transfer, where the embeddings for two languages are in alignment if word translations, e.g. *cat* and *Katze*, have similar representations (Mikolov et al., 2013a; Smith et al., 2017). Recently, large pre-trained models have largely subsumed word vectors based on their accuracy on downstream tasks, partly due to the fact that their word representations are context-dependent, allowing them to more richly capture the meaning of a word (Peters et al., 2018; Howard & Ruder, 2018; Radford et al., 2018; Devlin et al., 2018). Therefore, with the same goal of cross-lingual transfer but for these more complex models, we might consider contextual embedding alignment, where we observe whether word pairs within parallel sentences, e.g. *cat* in *"The cat sits"* and *Katze* in *"Die Katze sitzt,"* have similar representations.

One model relevant to these questions is multilingual BERT, a version of BERT pre-trained on 104 languages that achieves remarkable transfer on downstream tasks. For example, after the model is fine-tuned on the English MultiNLI training set, it achieves 74.3% accuracy on the test set in Spanish, which is only 7.1% lower than the English accuracy (Devlin et al., 2018; Conneau et al., 2018b). Furthermore, while the model transfers better to languages similar to English, it still achieves reasonable accuracies even on languages with different scripts.

However, given the way that multilingual BERT was pre-trained, it is unclear why we should expect such high zero-shot performance. Compared to monolingual BERT which exhibits no zero-shot transfer, multilingual BERT differs only in that (1) during pre-training (i.e. masked word prediction), each batch contains sentences from all of the languages, and (2) it uses a single shared vocabulary, formed by WordPiece on the concatenated monolingual corpora (Devlin et al., 2019). Therefore, we might wonder: (1) How can we better understand BERT's multilingualism? (2) Can we further improve BERT's cross-lingual transfer?

In this paper, we show that contextual embedding alignment is a useful concept for addressing these questions. First, we propose a contextual version of word retrieval to evaluate the degree of alignment, where a model is presented with two parallel corpora, and given a word within a sentence in one corpus, it must find the correct word and sentence in the other. Using this metric of alignment, we show that multilingual BERT achieves zero-shot transfer because its embeddings are partially aligned, as depicted in Figure 1, with the degree of alignment predicting the degree of downstream transfer.

Next, using between 10K and 250K sentences per language from the Europarl corpus as parallel data (Koehn, 2005), we propose a fine-tuning-based alignment procedure and show that it significantly improves BERT as a multilingual model. Specifically, on zero-shot XNLI, where the model is trained on English MultiNLI and tested on other languages (Conneau et al., 2018b), the aligned model improves accuracies by 2.78% on average over the base model, and it remarkably matches translate-train models for Bulgarian and Greek, which approximate the fully-supervised setting.

To put our results in the context of past work, we also use word retrieval to compare our fine-tuning procedure to two alternatives: (1) fastText augmented with sentence and aligned using rotations (Bojanowski et al., 2017; Rücklé et al., 2018; Artetxe et al., 2018), and (2) BERT aligned using rotations (Aldarmaki & Diab, 2019; Schuster et al., 2019; Wang et al., 2019). We find that when there are multiple occurences per word, fine-tuned BERT outperforms fastText, which outperforms rotation-aligned BERT. This result supports the intuition that contextual alignment is more difficult than its non-contextual counterpart, given that a rotation, at least when applied naively, is no longer sufficient to produce strong alignments. In addition, when there is only one occurrence per word, fine-tuned BERT matches the performance of fastText. Given that context disambiguation is no longer necessary, this result suggests that our fine-tuning procedure is able to align BERT at the type level to a degree that matches non-contextual approaches.

Finally, we use the contextual word retrieval task to conduct finer-grained analysis of multilingual BERT, with the goal of better understanding its strengths and shortcomings. Specifically, we find that base BERT has trouble aligning open-class compared to closed-class parts-of-speech, as well as word pairs that have large differences in usage frequency, suggesting insight into the pre-training procedure that we explore in Section 5. Together, these experiments support contextual alignment as an important task that provides useful insight into large multilingual pre-trained models.

## 2 RELATED WORK

**Word vector alignment.** There has been a long line of works that learn aligned word vectors from varying levels of supervision (Ruder et al., 2019). One popular family of methods starts with word vectors learned independently for each language (using a method like skip-gram with negative sampling (Mikolov et al., 2013b)), and it learns a mapping from source language vectors to target language vectors with a bilingual dictionary as supervision (Mikolov et al., 2013a; Smith et al., 2017; Artetxe et al., 2017). When the mapping is constrained to be an orthogonal linear transformation, the optimal mapping that minimizes distances between word pairs can be solved in closed form (Artetxe et al., 2016; Schonemann, 1966). Alignment is evaluated using bilingual lexicon induction, so these papers also propose ways to mitigate the hubness problem in nearest neighbors,

e.g. by using alternate similarity functions like CSLS (Conneau et al., 2018a). A recent set of works has also shown that the mapping can be learned with minimal to no supervision by starting with some minimal seed dictionary and alternating between learning the linear map and inducing the dictionary (Artetxe et al., 2018; Conneau et al., 2018a; Hoshen & Wolf, 2018; Xu et al., 2018; Chen & Cardie, 2018).

**Incorporating context into alignment.** One key challenge in making alignment context aware is that the embeddings are now different across multiple occurrences of the same word. Past papers have handled this issue by removing context and aligning the "average sense" of a word. In one such study, Schuster et al. (2019) learn a rotation to align contextual ELMo embeddings (Peters et al., 2018) with the goal of improving zero-shot multilingual dependency parsing, and they handle context by taking the average embedding for a word in all of its contexts. In another paper, Aldarmaki & Diab (2019) learn a rotation on sentence vectors, produced by taking the average word vector over the sentence, and they show that the resulting alignment also works well for word-level tasks. In a contemporaneous work, Wang et al. (2019) align not only the word but also the context by learning a linear transformation using word-aligned parallel data to align multilingual BERT, with the goal of improving zero-shot dependency parsing numbers. In this paper, we similarly align not only the word but also the context, and we also depart from these past works by using more expressive alignment methods than rotation.

**Incorporating parallel texts into pre-training.** Instead of performing alignment post-hoc, another line of works proposes contextual pre-training procedures that are more cross-lingually-aware. Wieting et al. (2019) pre-train sentence embeddings using parallel texts by maximizing similarity between sentence pairs while minimizing similarity with negative examples. Lample & Conneau (2019) propose a cross-lingual pre-training objective that incorporates parallel data in addition to monolingual corpora, leading to improved downstream cross-lingual transfer. In contrast, our method uses less parallel data and aligns existing pre-trained models rather than requiring pre-training from scratch.

**Analyzing multilingual BERT.** Pires et al. (2019) present a series of probing experiments to better understand multilingual BERT, and they find that transfer is possible even between dissimilar languages, but that it works better between languages that are typologically similar. They conclude that BERT is remarkably multilingual but falls short for certain language pairs.

## 3 METHODS

### 3.1 MULTILINGUAL PRE-TRAINING

We first briefly describe multilingual BERT (Devlin et al., 2018). Like monolingual BERT, multilingual BERT is pre-trained on sentences from Wikipedia to perform two tasks: masked word prediction, where it must predict words that are masked within a sentence, and next sentence prediction, where it must predict whether the second sentence follows the first one. The model is trained on 104 languages, with each batch containing training sentences from each language, and it uses a shared vocabulary formed by WordPiece on the 104 Wikipedias concatenated (Wu et al., 2016).

### 3.2 DEFINING AND EVALUATING CONTEXTUAL ALIGNMENT

In the following sections, we describe how to define, evaluate, and improve contextual alignment. Given two languages, a model is in *contextual alignment* if it has similar representations for word pairs within parallel sentences. More precisely, suppose we have $N$ parallel sentences $C = \{(\mathbf{s}^1, \mathbf{t}^1), ..., (\mathbf{s}^N, \mathbf{t}^N)\}$, where $(\mathbf{s}, \mathbf{t})$ is a source-target sentence pair. Also, let each sentence pair $(\mathbf{s}, \mathbf{t})$ have word pairs, denoted $a(\mathbf{s}, \mathbf{t}) = \{(i_1, j_1), ..., (i_m, j_m)\}$, containing position tuples $(i, j)$ such that the words $\mathbf{s}_i$ and $\mathbf{t}_j$ are translations of each other.[1] We will use $f$ to represent a pre-trained model such that $f(i, \mathbf{s})$ is the contextual embedding for the $i$th word in $\mathbf{s}$.

---

[1] These pairs are called word alignments in the machine translation community, but we use the term "word pairs" to avoid confusion with embedding alignment. Also, because BERT operates on subwords while the corpus is aligned at the word level, we keep only the BERT vector for the last subword of each word.

As an example, we might have the following sentence pair:

$$\mathbf{s} = \{\overset{0}{I} \overset{1}{ate} \overset{2}{the} \overset{3}{apple} \overset{4}{.}\} \quad \mathbf{t} = \{\overset{0}{Ich} \overset{1}{habe} \overset{2}{den} \overset{3}{Apfel} \overset{4}{gegessen} \overset{5}{.}\}$$
$$a(\mathbf{s}, \mathbf{t}) = \{(0,0), (1,4), (2,2), (3,3), (4,5)\}$$

Then, using the parallel corpus $C$, we can measure the contextual alignment of the model $f$ using its accuracy in *contextual word retrieval*. In this task, the model is presented with two parallel corpora, and given a word within a sentence in one corpus, it must find the correct word and sentence in the other. Specifically, we can define a nearest neighbor retrieval function

$$\text{neighbor}(i, \mathbf{s}; f, C) = \underset{\mathbf{t} \in C, \ 0 \leq j \leq \text{len}(\mathbf{t})}{\text{argmax}} \ \text{sim}(f(i, \mathbf{s}), f(j, \mathbf{t})),$$

where $i$ and $j$ denote the position within a sentence and sim is a similarity function. The accuracy is then given by the percentage of exact matches over the entire corpus, or

$$A(f; C) = \frac{1}{N} \sum_{(\mathbf{s},\mathbf{t}) \in C} \sum_{(i,j) \in a(\mathbf{s},\mathbf{t})} \mathbb{I}(\text{neighbor}(i, \mathbf{s}; f, C) = (j, \mathbf{t})),$$

where $\mathbb{I}$ represents the indicator function. We can perform the same procedure in the other direction, where we pull target words given source words, so we report the average between the two directions. As our similarity function, we use CSLS, a modified version of cosine similarity that mitigates the hubness problem, with neighborhood size 10 (Conneau et al., 2018a). One additional point is that this procedure can be made more or less contextual based on the corpus: a corpus with more occurrences for each word type requires better representations of context. Therefore, we also test non-contextual word retrieval by removing all but the first occurrence of each word type.

Given parallel data, these word pairs can be procured in an unsupervised fashion using standard techniques developed by the machine translation community (Brown et al., 1993). While these methods can be noisy, by running the algorithm in both the source-target and target-source directions and only keeping word pairs in their intersection, we can trade-off coverage for accuracy, producing a reasonably high-precision dataset (Och & Ney, 2003).

### 3.3 ALIGNING PRE-TRAINED CONTEXTUAL EMBEDDINGS

To improve the alignment of the model $f$ with respect to the corpus $C$, we can encapsulate alignment in the loss function

$$L(f; C) = - \sum_{(\mathbf{s},\mathbf{t}) \in C} \sum_{(i,j) \in a(\mathbf{s},\mathbf{t})} \text{sim}(f(i, \mathbf{s}), f(j, \mathbf{t})),$$

where we sum the similarities between word pairs. Because the CSLS metric is not easily optimized, we instead use the squared error loss, or $\text{sim}(f(i, \mathbf{s}), f(j, \mathbf{t})) = -||f(i, \mathbf{s}) - f(j, \mathbf{t})||_2^2$.

However, note that this loss function does not account for the informativity of $f$; for example, it is zero if $f$ is constant. Therefore, at a high level, we would like to minimize $L(f; C)$ while maintaining some aspect of $f$ that makes it useful, e.g. its high accuracy when fine-tuned on downstream tasks. Letting $f_0$ denote the initial pre-trained model before alignment, we achieve this goal by defining a regularization term

$$R(f; C) = \sum_{\mathbf{t} \in C} \sum_{i=1}^{\text{len}(\mathbf{t})} ||f(j, \mathbf{t}) - f_0(j, \mathbf{t})||_2^2,$$

which imposes a penalty if the target language embeddings stray from their initialization. Then, we sample minibatches $B \subset C$ and take gradient steps of the function $L(f; B) + \lambda R(f; B)$ directly on the weights of $f$, which moves the source embeddings toward the target embeddings while preventing the latter from drifting too far. In our experiments, we set $\lambda = 1$.

In the multilingual case, suppose we have $k$ parallel corpora $C^1, ..., C^k$, where each corpus has a different source language with the target language as English. Then, we sample equal-sized batches $B^i \subset C^i$ from each corpus and take gradient steps on $\sum_{i=1}^{k} L(f; B^i) + \lambda R(f; B^i)$, which moves all of the non-English embeddings toward English.

Note that this alignment method departs from prior work, in which each non-English language is rotated to match the English embedding space through individual learned matrices. Specifically, the most widely used post-hoc alignment method learns a rotation $W$ applied to the source vectors to minimize the distance between parallel word pairs, or

$$\min_{W} \sum_{(\mathbf{s},\mathbf{t}) \in C} \sum_{(i,j) \in a(\mathbf{s},\mathbf{t})} ||Wf(i,\mathbf{s}) - f(j,\mathbf{t})||_2^2 \quad s.t. \quad W^\top W = I. \tag{1}$$

This problem is known as the Procrustes problem and can be solved in closed form (Schonemann, 1966). This approach has the nice property that the vectors are only rotated, preserving distances and therefore the semantic information captured by the embeddings (Artetxe et al., 2016). However, rotation requires the strong assumption that the embedding spaces are roughly isometric (Søgaard et al., 2018), an assumption that may not hold for contextual pre-trained models because they represent more aspects of a word than just its type, i.e. context and syntax, which are less likely to be isomorphic between languages. Given that past work has also found independent alignment per language pair to be inferior to joint training (Heyman et al., 2019), another advantage of our method is that the alignment for all languages is done simultaneously.

As our dataset, we use the Europarl corpora for English paired with Bulgarian, German, Greek, Spanish, and French, the languages represented in both Europarl and XNLI (Koehn, 2005). After tokenization (Koehn et al., 2007), we produce word pairs using fastAlign and keep the one-to-one pairs in the intersection (Dyer et al., 2013). We use the most recent 1024 sentences as the test set, the previous 1024 sentences as the development set, and the following 250K sentences as the training set. Furthermore, we modify the test set accuracy calculation to only include word pairs not seen in the training set. We also remove any exact matches, e.g. punctuation and numbers, because BERT is already aligned for these pairs due to its shared vocabulary. Given that parallel data may be limited for low-resource language pairs, we also report numbers for 10K and 50K parallel sentences.

### 3.4    SENTENCE-AUGMENTED NON-CONTEXTUAL BASELINE

Given that there has been a long line of work on word vector alignment (Artetxe et al., 2018; Conneau et al., 2018a; Smith et al., 2017, *inter alia*), we also compare BERT to a sentence-augmented fastText baseline (Bojanowski et al., 2017). Following Artetxe et al. (2018), we first normalize, then mean-center, then normalize the word vectors, and we then learn a rotation with the same parallel data as in the contextual case, as described in Equation 1. We also strengthen this baseline by including sentence information: specifically, during word retrieval, we concatenate each word vector with a vector representing its sentence. Following Rücklé et al. (2018), we compute the sentence vector by concatenating the average, maximum, and minimum vector over all of the words in the sentence, a method that was shown to be state-of-the-art for a suite of cross-lingual tasks. We also experimented with other methods, such as first retrieving the sentence and then the word, but found this method resulted in the highest accuracy. As a result, the fastText vectors are 1200-dimensional, while the BERT vectors are 768-dimensional.

### 3.5    TESTING ZERO-SHOT TRANSFER

The next step is to determine whether better alignment improves cross-lingual transfer. As our downstream task, we use the XNLI dataset, where the English MultiNLI development and test sets are human-translated into multiple languages (Conneau et al., 2018b; Williams et al., 2018). Given a pair of sentences, the task is to predict whether the first sentence implies the second, where there are three labels: entailment, neutral, or contradiction. Starting from either the base or aligned multilingual BERT, we train on English and evaluate on Bulgarian, German, Greek, Spanish, and French, the XNLI languages represented in Europarl.

As our architecture, following Devlin et al. (2018), we apply a linear layer followed by softmax on the `[CLS]` embedding of the sentence pair, producing scores for each of the three labels. The model is trained using cross-entropy loss and selected based on its development set accuracy averaged across all of the languages. As a fully-supervised ceiling, we also compare to models trained and tested on the same language, where for the non-English training data, we use the machine translations of the English MultiNLI training data provided by Conneau et al. (2018b). While the quality of the training data is affected by the quality of the MT system, this comparison nevertheless serves as a good approximation for the fully-supervised setting.

|  | English | Bulgarian | German | Greek | Spanish | French | Average |
|---|---|---|---|---|---|---|---|
| **Translate-Train** | | | | | | | |
| Base BERT | 81.9 | 73.6 | 75.9 | 71.6 | 77.8 | 76.8 | 76.3 |
| **Zero-Shot**[a] | | | | | | | |
| Base BERT | 80.4 | 68.7 | 70.4 | 67.0 | 74.5 | 73.4 | 72.4 |
| Sentence-aligned BERT (rotation) | **81.1** | 68.9 | 71.2 | 66.7 | 74.9 | 73.5 | 72.7 |
| Word-aligned BERT (rotation) | 78.8 | 69.0 | 71.3 | 67.1 | 74.3 | 73.0 | 72.2 |
| Word-aligned BERT (fine-tuned) | 80.1 | **73.4** | **73.1** | **71.4** | **75.5** | **74.5** | **74.7** |
| XLM (MLM + TLM) | 85.0 | 77.4 | 77.8 | 76.6 | 78.9 | 78.7 | 79.1 |

Table 1: Accuracy on the XNLI test set, where we compare to base BERT (Devlin et al., 2018) and two rotation-based methods, sentence alignment (Aldarmaki & Diab, 2019) and word alignment (Wang et al., 2019). We also include the current state-of-the-art zero-shot achieved by XLM (Lample & Conneau, 2019). Rotation-based methods provide small gains on some languages but not others. On the other hand, after fine-tuning-based alignment, Bulgarian and Greek match the translate-train ceiling, while German, Spanish, and French close roughly one-third of the gap.

[a] Note that the zero-shot Base BERT numbers are slightly different from those reported in Devlin et al. (2019) because we select a single model using the average accuracy across the six languages. This selection method also accounts for the varying English accuracies across the zero-shot methods.

| Sentences | English | Bulgarian | German | Greek | Spanish | French | Average |
|---|---|---|---|---|---|---|---|
| None | 80.4 | 68.7 | 70.4 | 67.0 | 74.5 | 73.4 | 72.4 |
| 10K | 79.2 | 71.0 | 71.8 | 67.5 | 75.3 | 74.1 | 73.2 |
| 50K | **81.1** | 73 | 72.6 | 69.6 | 75 | **74.5** | 74.3 |
| 250K | 80.1 | **73.4** | **73.1** | **71.4** | **75.5** | **74.5** | **74.7** |

Table 2: Zero-shot accuracy on the XNLI test set, where we align BERT with varying amounts of parallel data. The method scales with the amount of data but achieves a large fraction of the gains with 50K sentences per language pair.

## 4 RESULTS

### 4.1 ZERO-SHOT XNLI TRANSFER

First, we test whether alignment improves multilingual BERT by applying the models to zero-shot XNLI, as displayed in Table 1. We see that our alignment procedure greatly improves accuracies, with all languages seeing a gain of at least $1\%$. In particular, the Bulgarian and Greek zero-shot numbers are boosted by almost $5\%$ each and match the translate-train numbers, suggesting that the alignment procedure is especially effective for languages that are initially difficult for BERT. We also run alignment for more distant language pairs (Chinese, Arabic, Urdu) and find similar results, which we report in the appendix.

Comparing to rotation-based methods (Aldarmaki & Diab, 2019; Wang et al., 2019), we find that a rotation produces small gains for some languages, namely Bulgarian, German, and Spanish, but is sub-optimal overall, providing evidence that the increased expressivity of our proposed procedure is beneficial for contextual alignment. We explore this comparison more in Section 5.1.

### 4.2 ALIGNMENT WITH LESS DATA

Given that our goal is zero-shot transfer, we cannot expect to always have large amounts of parallel data. Therefore, we also characterize the performance of our alignment method with varying amounts of data, as displayed in Table 2. We find that it improves transfer with as little as 10K sentences per language, making it a promising approach for low-resource languages.

|  | bg-en | de-en | el-en | es-en | fr-en | Average |
|---|---|---|---|---|---|---|
| **Contextual** |  |  |  |  |  |  |
| Aligned fastText + sentence | 44.0 | 46.4 | 42.0 | 48.6 | 44.5 | 45.1 |
| Base BERT | 19.5 | 26.1 | 13.9 | 32.5 | 28.3 | 24.1 |
| Word-aligned BERT (rotation) | 29.8 | 31.6 | 20.8 | 36.8 | 31.0 | 30.0 |
| Word-aligned BERT (fine-tuned) | **50.7** | **51.3** | **49.8** | **51.0** | **48.6** | **50.3** |
| **Non-Contextual** |  |  |  |  |  |  |
| Aligned fastText + sentence | 61.3 | **65.4** | 61.6 | **71.1** | 64.8 | 64.8 |
| Base BERT | 29.1 | 37.0 | 22.3 | 46.5 | 41.8 | 35.3 |
| Word-aligned BERT (rotation) | 39.6 | 43.6 | 32.4 | 51.4 | 46.1 | 42.6 |
| Word-aligned BERT (fine-tuned) | **62.8** | 64.3 | **67.5** | 68.4 | **66.3** | **65.9** |

Table 3: Word retrieval accuracy for the aligned sentence-augmented fastText baseline and BERT pre- and post-alignment. Across languages, base BERT has variable accuracy while fine-tuning-aligned BERT is consistently effective. Fine-tuned BERT also matches fastText in a version of the task where context is not necessary, suggesting that our method matches the type-level alignment of fastText while also aligning context.

# 5 ANALYSIS

## 5.1 WORD RETRIEVAL

In the following sections, we present word retrieval results to both compare our method to past work and better understand the strengths and weaknesses of multilingual BERT. Table 3 displays the word retrieval accuracies for the aligned sentence-augmented fastText baseline and BERT pre- and post-alignment. First, we find that in contextual retrieval, fine-tuned BERT outperforms fastText, which outperforms rotation-aligned BERT. This result supports the intuition that aligning large pre-trained models is more difficult than aligning word vectors, given that a rotation, at least when applied naively, produces sub-par alignments. In addition, fine-tuned BERT matches the performance of fastText in non-contextual retrieval, suggesting that our alignment procedure overcomes these challenges and achieves type-level alignment that matches non-contextual approaches. In the appendix, we also provide examples of aligned BERT disambiguating between different meanings of a word, giving qualitative evidence of the benefit of context alignment.

We also find that before alignment, BERT's performance varies greatly between languages, while after alignment it is consistently effective. In particular, Bulgarian and Greek initially have very low accuracies. This phenomenon is also reflected in the XNLI numbers (Table 1), where Bulgarian and Greek receive the largest boosts from alignment. Examining the connection between alignment and zero-shot more closely, we find that the word retrieval accuracies are highly correlated with downstream zero-shot performance (Figure 2), supporting our evaluation measure as predictive of cross-lingual transfer.

The language discrepancies are also consistent with a hypothesis by Pires et al. (2019) to explain BERT's multilingualism. They posit that due to the shared vocabulary, shared words between languages, e.g. numbers and names, are forced to have the same representation. Then, due to the masked word prediction task, other words that co-occur with these shared words also receive similar representations. If this hypothesis is true, then languages with higher lexical overlap with English are likely to experience higher transfer. As an extreme form of this phenomenon, Bulgarian and Greek have completely different scripts and should experience worse transfer than the common-script languages, an intuition that is confirmed by the word retrieval and XNLI accuracies. The fact that all languages are equally aligned with English post-alignment suggests that the pre-training procedure is suboptimal for these languages.

| Lexical Overlap | Numeral | Punctuation | Proper Noun | | | Average |
|---|---|---|---|---|---|---|
| Base BERT | 0.90 | 0.88 | 0.80 | | | 0.86 |
| Aligned BERT | 0.97 | 0.96 | 0.95 | | | 0.96 |
| Closed-Class | Determiner | Preposition | Conjunction | Pronoun | Auxiliary | Average |
| Base BERT | 0.76 | 0.72 | 0.71 | 0.70 | 0.61 | 0.70 |
| Aligned BERT | 0.91 | 0.86 | 0.89 | 0.89 | 0.84 | 0.88 |
| Open-Class | Noun | Adverb | Adjective | Verb | | Average |
| Base BERT | 0.61 | 0.57 | 0.50 | 0.49 | | 0.54 |
| Aligned BERT | 0.90 | 0.88 | 0.90 | 0.89 | | 0.89 |

Table 4: Accuracy by part-of-speech tag for non-contextual word retrieval. To achieve better word type coverage, we do not remove word pairs seen in the training set. The tags are grouped into lexically overlapping, closed-class, and open-class groups. The "Particle," "Symbol," "Interjection," and "Other" tags are omitted.

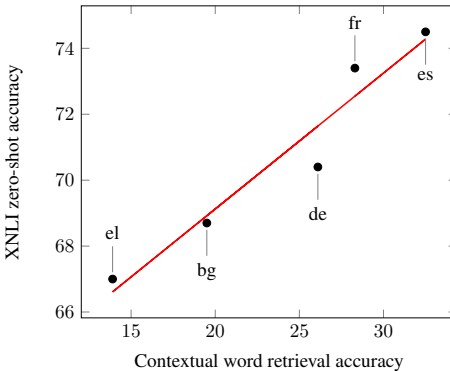

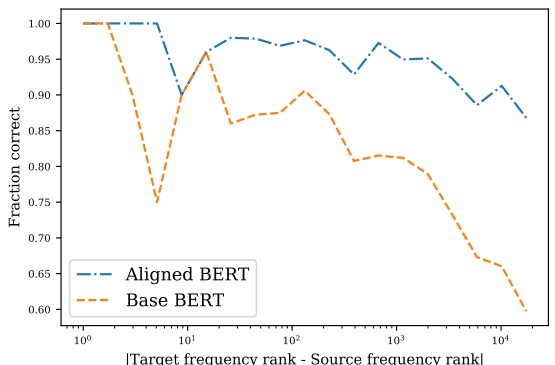

Figure 2: XNLI zero-shot versus word retrieval accuracy for base BERT, where each point is a language paired with English. This plot suggests that alignment correlates well with cross-lingual transfer.

Figure 3: Contextual word retrieval accuracy plotted against difference in frequency rank between source and target. The accuracy of base BERT plummets for larger differences, suggesting that its alignment depends on word pairs having similar usage statistics.

## 5.2 WORD RETRIEVAL PART-OF-SPEECH ANALYSIS

Next, to gain insight into the multilingual pre-training procedure, we analyze the accuracy broken down by part-of-speech using the Universal Part-of-Speech Tagset (Petrov et al., 2012), annotated using polyglot (Al-Rfou et al., 2013) for Bulgarian and spaCy (Honnibal & Montani, 2017) for the other languages, as displayed in Table 4. Unsurprisingly, multilingual BERT has high alignment out-of-the-box for groups with high lexical overlap, e.g. numerals, punctuation, and proper nouns, due to its shared vocabulary. We further divide the remaining tags into closed-class and open-class, where closed-class parts-of-speech correspond to fixed sets of words serving grammatical functions (e.g. determiner, preposition, conjunction, pronoun, and auxiliary), while open-class parts-of-speech correspond to lexical words (e.g. noun, adverb, adjective, verb). Interestingly, we see that base BERT has consistently lower accuracy for closed-class versus open-class categories ($0.70$ vs $0.54$), but that this discrepancy disappears after alignment ($0.88$ vs $0.89$).

## 5.3 USAGE HYPOTHESIS FOR ALIGNMENT

From this closed-class vs open-class difference, we hypothesize that BERT's alignment of a particular word pair is influenced by the similarity of their usage statistics. Specifically, given that BERT is trained through masked word prediction, its embeddings are in large part determined by

the co-occurrences between words. Therefore, two words that are used in similar contexts should be better aligned. This hypothesis provides an explanation of the closed-class vs open-class difference: closed-class words are typically grammatical, so they are used in similar ways across typologically similar languages. Furthermore, these words cannot be substituted for one another due to their grammatical function. Therefore, their usage statistics are a strong signature that can be used for alignment. On the other hand, open-class words can be substituted for one another: for example, in most sentences, the noun tokens could be replaced by a wide range of semantically dissimilar nouns with the sentence remaining syntactically well-formed. By this effect, many nouns have similar co-occurrences, making them difficult to align through masked word prediction alone.

To further test this hypothesis, we plot the word retrieval accuracy versus the difference between the frequency rank of the target and source word, where this difference measures discrepancies in usage, as depicted in Figure 3. We see that accuracy drops off significantly as the source-target difference increases, supporting our hypothesis. Furthermore, this shortcoming is remedied by alignment, revealing another systematic deficiency of multilingual pre-training.

## 6 CONCLUSION

Given that the degree of alignment is causally predictive of downstream cross-lingual transfer, contextual alignment proves to be a useful concept for understanding and improving multilingual pre-trained models. Given small amounts of parallel data, our alignment procedure improves multilingual BERT and corrects many of its systematic deficiencies. Contextual word retrieval also provides useful new insights into the pre-training procedure, opening up new avenues for analysis.

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

|  | English | Bulgarian | German | Greek | Spanish | French | Arabic | Chinese | Urdu | Average |
|---|---|---|---|---|---|---|---|---|---|---|
| Translate-Train |  |  |  |  |  |  |  |  |  |  |
| Base BERT | 81.9 | 73.6 | 75.9 | 71.6 | 77.8 | 76.8 | 70.7 | 76.6 | 61.6 | 74.1 |
| Zero-Shot |  |  |  |  |  |  |  |  |  |  |
| Base BERT | 80.4 | 68.7 | 70.4 | 67.0 | 74.5 | 73.4 | 65.6 | 70.6 | 60.3 | 70.1 |
| Aligned BERT (20K sent) | **80.8** | **71.6** | **72.5** | **68.1** | **74.7** | **73.6** | **66.3** | **71.5** | **61.1** | **71.1** |

Table 5: Zero-shot accuracy on the XNLI test set with more languages, where we use 20K parallel sentences for each language paired with English. This result confirms that the alignment method works for distant languages and a variety of parallel corpora, including Europarl, MultiUN, and Tanzil, which contains sentences from the Quran (Koehn, 2005; Eisele & Chen, 2010; Tiedemann, 2012).

# A   APPENDIX

## A.1   OPTIMIZATION HYPERPARAMETERS

For both alignment and XNLI optimization, we use a learning rate of $5 \times 10^{-5}$ with Adam hyperparameters $\beta = (0.9, 0.98)$, $\epsilon = 10^{-9}$ and linear learning rate warmup for the first 10% of the training data. For alignment, the model is trained for one epoch, with each batch containing 2 sentence pairs per language. For XNLI, each model is trained for 3 epochs with 32 examples per batch, and 10% dropout is applied to the BERT embeddings.

## A.2   ALIGNMENT OF CHINESE, ARABIC, AND URDU

In Table 5, we report numbers for additional languages, where we align a single BERT model for all eight languages and then fine-tune on XNLI. We use 20K sentences per language, where we use the MultiUN corpus for Arabic and Chinese (Eisele & Chen, 2010), the Tanzil corpus for Urdu (Tiedemann, 2012), and the Europarl corpus for the other five languages (Koehn, 2005). This result confirms that the alignment method works for a variety of languages and corpora. Furthermore, the Tanzil corpus consists of sentences from the Quran, suggesting that the method works even when the parallel corpus and downstream task contain sentences from entirely different domains.

## A.3   EXAMPLES OF CONTEXT-AWARE RETRIEVAL

In this section, we qualitatively show that aligned BERT is able to disambiguate between different occurences of a word.

First, we find two meanings of the word "like" occurring in the English-German Europarl test set. Note also that in the second and third example, the two senses of "like" occur in the same sentence.

- This empire did not look for colonies far from home or overseas, **like** most Western European States, but close by.
  Dieses Reich suchte seine Kolonien nicht weit von zu Hause und in bersee **wie** die meisten westeuropäischen Staaten, sondern in der unmittelbaren Umgebung.

- **Like** other speakers, I would like to support the call for the arms embargo to remain.
  **Wie** andere Sprecher, so möchte auch ich den Aufruf zur Aufrechterhaltung des Waffenembargos untersttzen.

- Like other speakers, I would **like** to support the call for the arms embargo to remain.
  Wie andere Sprecher, so **möchte** auch ich den Aufruf zur Aufrechterhaltung des Waffenembargos untersttzen.

- I would also **like**, although they are absent, to mention the Commission and the Council.
  Ich **möchte** mir sogar erlauben, die Kommission und den Rat zu nennen, auch wenn sie nicht anwesend sind.

Multiple meanings of "order":

- Moreover, the national political elite had to make a detour in Ambon in **order** to reach the civil governor's residence by warship.

  In Ambon mußte die politische Spitze des Landes auch noch einen Umweg machen, **um** mit einem Kriegsschiff die Residenz des Provinzgouverneurs zu erreichen.

- Although the European Union has an interest in being surrounded by large, stable regions, the tools it has available in **order** to achieve this are still very limited.

  Der Europäischen Union ist zwar an großen stabilen Regionen in ihrer Umgebung gelegen, aber sie verfgt nach wie vor nur ber recht begrenzte Instrumente, **um** das zu erreichen.

- We could reasonably expect the new Indonesian government to take action in three fundamental areas: restoring public **order**, prosecuting and punishing those who have blood on their hands and entering into a political dialogue with the opposition.

  Von der neuen indonesischen Regierung darf man mit Fug und Recht drei elementare Maßnahmen erwarten: die Wiederherstellung der öffentlichen **Ordnung**, die Verfolgung und Bestrafung derjenigen, an deren Händen Blut klebt, und die Aufnahme des politischen Dialogs mit den Gegnern.

- Firstly, I might mention the fact that the army needs to be reformed, secondly that a stable system of law and **order** needs to be introduced.

  Ich nenne hier an erster Stelle die notwendige Reform der Armee, ferner die Einfhrung eines stabilen Systems rechtsstaatlicher **Ordnung**.

Multiple meanings of "support":

- Financial **support** is needed to enable poor countries to take part in these court activities.

  Arme Länder müssen finanziell **unterstützt** werden, damit auch sie sich an der Arbeit des Gerichtshofs beteiligen können.

- We must help them and ensure that a proper action plan is implemented to **support** their work.

  Es gilt einen wirklichen Aktionsplan auf den Weg zu bringen, um die Arbeit dieser Organisationen zu **unterstützen**.

- So I hope that you will all **support** this resolution condemning the abominable conditions of prisoners and civilians in Djibouti.

  Ich hoffe daher, daß Sie alle diese Entschließung **befürworten**, die die entsetzlichen Bedingungen von Inhaftierten und Zivilpersonen in Dschibuti verurteilt.

- It would be difficult to **support** a subsidy scheme that channelled most of the aid to the large farms in the best agricultural regions.

  Es wäre auch problematisch, ein Beihilfesystem zu **befürworten**, das die meisten Beihilfen in die großen Betriebe in den besten landwirtschaftlichen Gebieten lenkt.

Multiple meanings of "close":

- This empire did not look for colonies far from home or overseas, like most Western European States, but **close** by.

  Dieses Reich suchte seine Kolonien nicht weit von zu Hause und in bersee wie die meisten westeuropäischen Staaten, sondern in der unmittelbaren **Umgebung**.

- In addition, if we are to shut down or refuse investment from every company which may have an association with the arms industry, then we would have to **close** virtually every American and Japanese software company on the island of Ireland with catastrophic consequences.

  Wenn wir zudem jedes Unternehmen, das auf irgendeine Weise mit der Rstungsindustrie verbunden ist, schließen oder Investitionen dieser Unternehmen unterbinden, dann mßten wir so ziemlich alle amerikanischen und japanischen Softwareunternehmen auf der irischen Insel **schließen**, was katastrophale Auswirkungen hätte.

- On the other hand, the deployment of resources left over in the Structural Funds from the programme planning period 1994 to 1999 is hardly worth considering as the available funds have already been allocated to specific measures, in this case in **close** collaboration with the relevant French authorities.

  Die Verwendung verbliebener Mittel der Strukturfonds aus dem Programmplanungszeitraum 1994 bis 1999 ist dagegen kaum in Erwägung zu ziehen, da die verfgbaren Mittel bereits bestimmten Maßnahmen zugewiesen sind, und zwar im konkreten Fall im **engen** Zusammenwirken mit den zuständigen französischen Behörden.

- This is particularly justified given that, as already stated, many Member States have very **close** relations with Djibouti.

  Zumal, wie erwähnt, viele Mitgliedstaaten sehr **enge** Beziehungen zu Dschibuti unterhalten.

- Mr President, it is regrettable that, at the **close** of the 20th century, a century symbolised so positively by the peaceful women's revolution, there are still countries, such as Kuwait and Afghanistan, where half the population, women that is, is still denied fundamental human rights.

  Herr Präsident! Es ist wirklich bedauerlich, daß es am **Ende** des 20. Jahrhunderts, eines so positiv von der friedlichen Revolution der Frauen geprägten Jahrhunderts, noch immer Länder wie Kuwait und Afghanistan gibt, in denen der Hälfte der Bevölkerung, den Frauen, die elementaren Menschenrechte verweigert werden.

