# OpenReview forum: "Multilingual Alignment of Contextual Word Representations"
_ICLR.cc/2020/Conference — Accept (Poster)_

### Official Review · AnonReviewer3 · 2019-10-22
**Official Blind Review #3**

**Rating:** 6

**Review:**

This paper conducts a series of experiments on the multilingual BERT model of Devlin et al., aiming to inject stronger bilingual knowledge into the model for improved 'Aligned BERT'. The knowledge originating from parallel (Europarl) data improves the model significantly as shown on tasks such as contextual and non-contextual word retrieval as well as in zero-shot XNLI task. The paper continues the line of work on cross-lingual contextualised word embeddings, and it brings several minor contributions, but overall I do not see it as a very inspiring piece of work, and it leaves open several very important questions, in particular its relationship to prior work and some potentially stronger baselines than the ones reported in the paper, plus more experiments with more distant language pairs.

I am not exactly sure that the comparison between 'Aligned BERT' and the main baseline 'Aligned fastText + sentence' is completely fair. 'Aligned BERT' uses more than 2M Europarl sentences to learn the alignment, while the standard alignment methods for learning cross-lingual word embeddings (see e.g. Ruder et al.'s survey) typically rely only on 5k translation pairs or even less pairs. There is a huge difference in the strength of the bilingual signal between 2M parallel sentences and, say, 2k, word translation pairs.

The main goal of the paper is to improve alignment of the starting multilingual BERT model, but I wonder why the authors have not compared to a more suitable XLM baseline of Lample and Conneau (NeurIPS 2019; the paper has been on arXiv since January 2019) - the XLM model uses exactly the same resources as 'Aligned BERT': parallel sentences from Europarl, while the main baseline here uses only seed dictionaries to learn the mapping. Regarding the baselines, it is also not clear to me why the authors have not compared to previous work of Schuster et al. (2019) and Aldarmaki and Diab (2019) at least in tasks where the models can be directly compared (XNLI or non-contextual word retrieval). Also, another non-contextual model which is worth trying is a joint model which relies on parallel sentences (similar to Ormazabal et al., ACL-19).

For the 'Aligned fastText + sentence' baseline, it would be interesting to report numbers with another (hybrid) baseline model that combines aligned fastText vectors with sentence encodings produced by multilingual BERT or some other multilingual sentence encoder (such as LASER, see Schwenk et al., 2019). Simply taking min, max, and avg vectors over all the sentence words might not be the best way to encode the sentence, and I would like to see more experiments here.

The paper makes some claims on novelty which 1) partially overlap with prior work, or 2) it does not cite related work while it leans on its findings. For instance on Page 4, the authors claim that their "(...) alignment method departs from prior work, in which each non-English language is rotated to match the English embedding space through individual learned matrices." However, there is at least one previous paper (Heyman et al., NAACL 2019) which did the same thing as the authors and showed that departing from learning projections only to English leads to more robust multilingual embeddings. Further, also on Page 4, the authors discuss that the assumption on learning good rotation matrices relies on the assumption of rough/approximate isomorphism without citing a body of related work that actually investigated this assumption such as the work of Sogaard et al. (ACL 2018). Also, the paper should do a better job in Section 2 and cover "word vector alignment" in more detail (e.g., a good starting point might be Ruder et al.'s survey paper on cross-lingual word embeddings).

The assumption of rough/approximate isomorphism is problematic also for non-contextual cross-lingual embeddings in settings with more distant language pairs. The authors mention that it may not hold for 'contextual pre-trained models given their increased complexity'. This is very imprecise writing taking place imho: 1) it is not clear why it should not hold in the case of contextual pre-trained models (at least for similar languages). Are there any properties of the contextual models that invalidate that assumption? It is also not exactly shown why contextual pre-trained models have increased complexity compared to e.g. fastText. How does one measure that 'model complexity' in objective terms? In fact, the paper would contribute immensely from more precise writing: e.g., on Page 3 contextual alignment of the model f is defined as accuracy in contextual word retrieval. This reads as defining a critical concept or a task as an evaluation measure (that measures the success of that task). In Introduction, the paper aims to "better understand BERT’s multilingualism", but I do not see how it contributes to our better understanding of BERT's multilingualism besides a pretty straightforward claim that it shows less multilingual potential when doing experiments with Greek and Bulgarian that use different scripts. Figure 2 and Figure 3 also do not bring anything new - the paper seems to just state known facts without proposing new solutions on how to e.g. learn better alignments for Greek or Bulgarian.

One important analysis aspect is missing from the paper: there are no experiments with more distant language pairs (the most distant language pair is English-Greek). I would like to see more experiments in this space. Another experiment which would contribute to the paper is the analysis of the importance of parallel corpora size. How much does the model lose in its performance by shrinking the parallel corpus? We cannot expect having 2M sentences for so many language pairs, and, even if we do have the data, the paper does not convince me that I should use 'Aligned BERT' instead of e.g. the XLM model of Lample and Conneau.

Minor remarks:
As a variant of the contextual word retrieval, have the authors tested if a correct target language sentence can be retrieved only looking at the context of the source language word? This would provide some insight on the importance of modeling context via BERT versus via simple context averaging.

Regarding the analysis between closed-class and open-class words performance, the difference in performance can be due to mere frequency: closed-class word types are very scarce, but their corpus frequency is quite high which also leads to learning better representations in the first place, as well as better alignments later on.

**Experience Assessment:**

I have published one or two papers in this area.

**Review Assessment: Checking Correctness Of Derivations And Theory:**

I assessed the sensibility of the derivations and theory.

**Review Assessment: Checking Correctness Of Experiments:**

I carefully checked the experiments.

**Review Assessment: Thoroughness In Paper Reading:**

I read the paper thoroughly.

---

> ### Author Response · Authors · 2019-11-10
> **Thank you for the thorough and insightful response (part 1)**
>
> Thank you for your thorough and insightful response, which we have found very useful in improving the paper. We will respond to each comment in-line.
>
> > I am not exactly sure that the comparison between 'Aligned BERT' and the main baseline 'Aligned fastText + sentence' is completely fair. 'Aligned BERT' uses more than 2M Europarl sentences to learn the alignment, while the standard alignment methods for learning cross-lingual word embeddings (see e.g. Ruder et al.'s survey) typically rely only on 5k translation pairs or even less pairs. There is a huge difference in the strength of the bilingual signal between 2M parallel sentences and, say, 2k, word translation pairs.
>
> Thank you for making this point; the data requirements of the method are indeed important considering that the goal is zero-shot transfer. We have run our method using 10K, 50K, and 250K sentences per language pair, and the results are in Table 2 of the revised submission. The result is that the method produces large gains even with 10K sentences. We would also like to point out that (1) the reported numbers in our original submission use 250K sentences, and (2) we use the same level of supervision for our aligned fastText method. We have edited the paper to make these two points clearer.
>
> > I wonder why the authors have not compared to a more suitable XLM baseline of Lample and Conneau (NeurIPS 2019; the paper has been on arXiv since January 2019) - the XLM model uses exactly the same resources as 'Aligned BERT'
>
> We agree that XLM uses parallel data in a similar way to our paper and achieves impressive XNLI numbers, so we have added it as a point of comparison in Table 1. However, we would like to note our method uses much less supervision than XLM: the numbers reported in our original submission use 250k sentences, and the method also works with 10k sentences. Our method also has the advantage that it can be performed in a day with a single GPU, whereas pre-training from scratch requires compute resources that are typically available only at large companies. If new parallel data becomes available, our method can also be quickly applied to take advantage of it. It’s also the case that the XLM numbers are not completely comparable to those of BERT: their MLM model, which uses no parallel sentences, still outperforms BERT on English XNLI and overall, as shown in Table 1 of their paper. The purpose of our paper was to explore and analyze how the idea of embedding alignment can be applied to contextual pre-trained models, so we found it more meaningful to perform controlled experiments to tease out the benefits of specific techniques, with BERT as a representative pre-trained model. However, we do agree that XLM is of interest and hope to explore the application of our method to the model in future work.
>
> > Regarding the baselines, it is also not clear to me why the authors have not compared to previous work of Schuster et al. (2019) and Aldarmaki and Diab (2019) at least in tasks where the models can be directly compared (XNLI or non-contextual word retrieval).
>
> Thank you for pointing this out. We agree that the paper would benefit from more comparisons to existing methods. Therefore, we have added two comparisons that are quick to implement and most directly comparable to our method: (1) the method from Aldarmaki and Diab (2019), which aligns sentence vectors using a linear transformation, and (2) the contemporaneous method from Wang et al. (EMNLP 2019), which aligns word pairs within parallel sentences using a linear transformation. The results suggest that a linear transformation is suboptimal for producing strong alignments, as displayed in Tables 1 and 3.
>
> > For the 'Aligned fastText + sentence' baseline, it would be interesting to report numbers with another (hybrid) baseline model that combines aligned fastText vectors with sentence encodings produced by multilingual BERT or some other multilingual sentence encoder (such as LASER, see Schwenk et al., 2019). Simply taking min, max, and avg vectors over all the sentence words might not be the best way to encode the sentence, and I would like to see more experiments here.
>
> Thank you for mentioning these works. We agree that more experiments with other sentence encoders could provide more insight, which we would like to experiment with in the future. Also, we would like to note that appending the min, max, and avg was shown to be state-of-the-art for cross-lingual tasks over more complex sentence encoders, so we do believe it is a competitive method (Rücklé et al., 2018). We also chose this method over other more complex sentence encoders because we wanted to ask the question, "What is the best we can do with non-contextual word vectors?", as a direct comparison to contextual word vectors.

---

> ### Author Response · Authors · 2019-11-10
> **Thank you for the thorough and insightful response (part 2)**
>
> > The paper makes some claims on novelty which 1) partially overlap with prior work, or 2) it does not cite related work while it leans on its findings ... Also, the paper should do a better job in Section 2 and cover "word vector alignment" in more detail
>
> Thank you for providing these references. We have edited to manuscript to cite them accordingly and improved Section 2. But on the comment about novelty, we would like to note that (1) to the best of our knowledge, our departure from linear transformations has not been done in prior work, and (2) it is not obvious without experimental study whether results about non-contextual word vectors transfer to large pre-trained models.
>
> > The authors mention that it may not hold for 'contextual pre-trained models given their increased complexity'. This is very imprecise writing taking place ... In fact, the paper would contribute immensely from more precise writing: e.g., on Page 3 contextual alignment of the model f is defined as accuracy in contextual word retrieval. This reads as defining a critical concept or a task as an evaluation measure (that measures the success of that task).
>
> Thanks for pointing this out. What we meant to say was, contextual word vectors must contain much more information than non-contextual ones, including the word, its context, syntax, and more. We hypothesize that this fact makes it much less likely that two languages will be isomorphic, because the word, context, and syntax must all be isomorphically represented in the vector space. We also agree with your comment on the definition of alignment and have modified the paper accordingly. If there are more areas in the revision that you find imprecise, we would be happy to make further changes.
>
> > The paper aims to "better understand BERT’s multilingualism", but I do not see how it contributes to our better understanding of BERT's multilingualism besides a pretty straightforward claim that it shows less multilingual potential when doing experiments with Greek and Bulgarian that use different scripts. Figure 2 and Figure 3 also do not bring anything new.
>
> To the best of our knowledge, Figure 2 and Figure 3 are not known in the literature. Figure 2 provides a precise way of measuring the alignment of a contextual pre-trained model, and it shows that this evaluation measure correlates very well with downstream zero-shot transfer. As far as we know, the evaluation measure and the correlation results are new. Figure 3 uses our evaluation measure to show that BERT is better aligned when the two words have similar usage frequencies. Also, Table 4 shows differences between open-class and closed-class parts-of-speech. Given that multilingual BERT is still not very well understood, we believe that these findings provide useful insights into the model.
>
> > The paper seems to just state known facts without proposing new solutions on how to e.g. learn better alignments for Greek or Bulgarian.
>
> We present our alignment method as a solution to these deficiencies: it closes the gap between Greek/Bulgarian and German/Spanish/French, open and closed-class parts-of-speech, and word pairs with different frequencies.
>
> > One important analysis aspect is missing from the paper: there are no experiments with more distant language pairs (the most distant language pair is English-Greek). I would like to see more experiments in this space.
>
> This is a good point and we hope to run our method on more language pairs, especially distant ones, in the future.
>
> > Another experiment which would contribute to the paper is the analysis of the importance of parallel corpora size. How much does the model lose in its performance by shrinking the parallel corpus? We cannot expect having 2M sentences for so many language pairs, and, even if we do have the data, the paper does not convince me that I should use 'Aligned BERT' instead of e.g. the XLM model of Lample and Conneau.
>
> Thank you for pointing this out; we agree that we cannot expect to have 2M sentences for so many language pairs. As mentioned in part 1 of this comment, we use 250K sentences per pair. Also, we present new experiments for 10K and 50K sentences. Given that the alignment procedure is lightweight and can be applied to any existing pre-trained model, we envision our alignment procedure to be useful for the (104 - 15) languages that BERT was trained on but not XLM. We give further advantages of our method above in our response. Finally, while we believe that our method is practically useful, the focus of our paper is to show how embedding alignment can be applied to pre-trained models, rather than to present a new state-of-the-art.

---

> > ### Comment · AnonReviewer3 · 2019-11-12
> > **Thanks for the responses!**
> >
> > I would like to thank the authors for a very substantial set of responses to my questions. While I'd still like to see more language pairs and additional experiments already in this work (and not as future work), I believe that the revisions and edits have made this submission stronger, and I'm fine with raising my score accordingly. After all, I believe that the community working on cross-lingual representations and xling transfer will find this work valuable, at least as a reference point.
> >
> > Why do two "rotation" baselines show such bad results? Do you have any explanation for that?
> >
> > Also, if I'm not mistaken, it seems that performance with your method saturates after relying on >250K parallel sentence, but that is not the case for XLM. How can we leverage richer/larger parallel corpora with your method?

---

> > > ### Author Response · Authors · 2019-11-15
> > > **Thanks again for the feedback.**
> > >
> > > Thanks again for the feedback. As an update, we have run our method on more distant languages (Chinese, Arabic, and Urdu), where we align a single BERT model on all eight languages (the five Europarl languages and the three additional ones) using 20K sentences per language. We see similar gains for the new languages, and the results are in the appendix. Interestingly, the Urdu parallel sentences come from the Quran, which is a very different domain from XNLI. Therefore, given that the Bible parallel corpus exists for 100 languages, it may be feasible to perform alignment for many languages.
> > >
> > > > Why do two "rotation" baselines show such bad results? Do you have any explanation for that?
> > >
> > > Our hypothesis is that rotation is not expressive enough, so it leads to better word retrieval results and German XNLI accuracy but lackluster results otherwise. Past work using rotations to align contextual word embeddings focus on dependency parsing (Schuster et al., 2019, Wang et al., 2019), which is word-level rather than sentence-level and more syntactic than semantic compared to XNLI. Therefore, perhaps there are key differences between the tasks that make rotation sufficient for one but not the other. We do think this discrepancy deserves further experimentation.
> > >
> > > > Also, if I'm not mistaken, it seems that performance with your method saturates after relying on >250K parallel sentence, but that is not the case for XLM. How can we leverage richer/larger parallel corpora with your method?
> > >
> > > While we do not have experiments using >250K sentences, we agree that accuracy gains seem to saturate as we increase the amount of parallel data. One axis along which we might modify the method would be to make it stricter (e.g. enforce a squared error loss on all of the layers of BERT, rather than just the last layer), or looser (e.g. encourage a shared embedding space without enforcing closeness in L2 norm, like the translation modeling objective in XLM). Our intuition is that stricter methods are more data efficient, but looser methods might be able to produce higher numbers given more data because they are less restrictive. It would be interesting to explore this hypothesis.

---

### Official Review · AnonReviewer2 · 2019-10-23
**Official Blind Review #2**

**Rating:** 6

**Review:**

This paper presents a new method to further align multilingual BERT by learning a transformation to minimize distances in a parallel corpus.

I think that this is overall a solid work. Although simple, the proposed method is well-motivated, and the reported results are generally convincing. However, I think that the paper lacks an appropriate comparison with similar methods in the literature, and the separation between the real evaluation in a downstream task (XNLI) and the analysis on a rather artificial contextual word retrieval task (which favors the proposed system) is not clear enough.

More concretely, these are the aspects that I think the paper could (and should) improve:

- You are not comparing to any baseline using parallel data with contextual embeddings. You should at least compare your method to Schuster et al. (2019) and/or Aldarmaki & Diab (2019), who further align multilingual BERT in a supervised manner as you do, as well as Lample and Conneau (2019), who propose an alternative method to leverage parallel data during the training of multilingual BERT. In fact, while you do improve over multilingual BERT, your results in XNLI are far from the current state-of-the-art, and this is not even mentioned in the paper.

- The "contextual word retrieval" task you propose is rather artificial and lacks any practical interest. It is not surprising that your proposed method is strong at it, as this is essentially how you train it (you are even using different subsets of the exact same corpus for train/test). The task is still interesting for analysis -which is in fact one of the main strengths of the paper- but it should be presented as such. Please consider restructuring your paper and moving all these results to the analysis section, where they really belong.

- I do not see the point of the "non-contextual word retrieval" task, when you are in fact using the context (the fact that there is only one occurrence per word type doesn't change that). This task is even more artificial than the "contextual word retrieval" one. Again, it can have some interest as part of the analysis (showing that the gap between aligned fasttext and aligned BERT goes down from table 1 to table 2), but presenting it as a separate task as if it had some value on its own looks wrong. From my point of view, the real "non-contextual word retrieval" task would be bilingual lexicon induction (i.e. dictionary induction), which is more interesting as a task (as the induced dictionaries can have practical applications) and has been widely studied in the literature.

- I really dislike the statement that contextual methods are "unequivocally better than non-contextual methods for multilingual tasks" on the basis of the non-contextual word retrieval results. If you want to make such a strong statement, you should at least show that your method is better than non-contextual ones in a task where the latter are known to be strong (i.e. bilingual lexicon induction, see above). However, your comparison is limited to a new task you introduce that clearly favors your own method, and in fact requires using the non-contextual methods in a non-standard way (concatenating the word embeddings with the avg/max/min sentence embeddings). Please either remove this statement or run a fair comparison in bilingual lexicon induction (and preferably do both).

- BERT works at the subword level but, from what I understand, your parallel corpus (both for train/test) is aligned at the word level. It is not clear at all how this mismatch in the tokenization is handled.


Minor details that did not influence my score:

- Calling "fully-supervised" to the "translate-train" system is misleading. Please simply call it "translate-train".

- I assume you want to refer to Figure 3 instead of Figure 2 in Section 5.2

**Experience Assessment:**

I have published in this field for several years.

**Review Assessment: Checking Correctness Of Derivations And Theory:**

N/A

**Review Assessment: Checking Correctness Of Experiments:**

I carefully checked the experiments.

**Review Assessment: Thoroughness In Paper Reading:**

I read the paper thoroughly.

---

> ### Author Response · Authors · 2019-11-10
> **Thank you for the insightful feedback.**
>
> Thank you for the insightful feedback. We have incorporated them into the revision, and we address the comments below in-line:
>
> > You are not comparing to any baseline using parallel data with contextual embeddings. You should at least compare your method to Schuster et al. (2019) and/or Aldarmaki & Diab (2019), who further align multilingual BERT in a supervised manner as you do, as well as Lample and Conneau (2019), who propose an alternative method to leverage parallel data during the training of multilingual BERT. In fact, while you do improve over multilingual BERT, your results in XNLI are far from the current state-of-the-art, and this is not even mentioned in the paper.
>
> Thank you for this comment. We have added two comparisons that are quick to implement and most directly comparable to our method: (1) the method from Aldarmaki and Diab (2019), which aligns sentence vectors using a linear transformation, and (2) the contemporaneous method from Wang et al. (EMNLP 2019), which aligns word pairs within parallel sentences using a linear transformation. In terms of comparing to the method in Lample and Conneau (2019), we do not have the compute to pre-train from scratch. Also, the numbers in their paper are not directly comparable to BERT, given that their MLM model, which uses no parallel data, still outperforms BERT on English XNLI and overall. Nonetheless, we have included their numbers in the XNLI table to represent the state-of-the-art.
>
> > The "contextual word retrieval" task you propose is rather artificial and lacks any practical interest. It is not surprising that your proposed method is strong at it, as this is essentially how you train it (you are even using different subsets of the exact same corpus for train/test). The task is still interesting for analysis -which is in fact one of the main strengths of the paper- but it should be presented as such. Please consider restructuring your paper and moving all these results to the analysis section, where they really belong.
>
> This is a good point. We completely agree and have restructured the paper accordingly.
>
> > I do not see the point of the "non-contextual word retrieval" task, when you are in fact using the context (the fact that there is only one occurrence per word type doesn't change that). This task is even more artificial than the "contextual word retrieval" one. Again, it can have some interest as part of the analysis (showing that the gap between aligned fasttext and aligned BERT goes down from table 1 to table 2), but presenting it as a separate task as if it had some value on its own looks wrong. From my point of view, the real "non-contextual word retrieval" task would be bilingual lexicon induction (i.e. dictionary induction), which is more interesting as a task (as the induced dictionaries can have practical applications) and has been widely studied in the literature.
>
> We agree that it would indeed be ideal to use BLI instead of non-contextual word retrieval. However, given that BERT needs the context as well as the word itself to produce a vector, non-contextual word retrieval is an easy way to produce bilingual dictionaries with sentences attached. Of course, it is not realistic to expect the source and target sentences to be parallel as well, which makes BLI a harder task for BERT. But, as you mention, we do think that the task has value for analysis because it can be accomplished without any representation of context. In particular, in the original contextual word retrieval task, BERT could be outperforming fastText for two reasons: (1) it can better represent context, and (2) it is better aligned. The point of introducing this task was to reduce the contribution of (1) and more directly compare the alignment between the two models. We have modified the framing in the paper to make this intention clearer.
>
> > I really dislike the statement ...
>
> We agree that this statement is not well-supported, so we have removed it from the paper. As described above, we have modified the framing and claims about non-contextual word retrieval.
>
> > BERT works at the subword level but, from what I understand, your parallel corpus (both for train/test) is aligned at the word level.
>
> Thank you for noticing this point, which was not addressed in the paper. We handle this issue by keeping the vector for the last subword of each word, and we have added a sentence to this effect in the paper.
>
> > Calling "fully-supervised" to the "translate-train" system is misleading. Please simply call it "translate-train". I assume you want to refer to Figure 3 instead of Figure 2 in Section 5.2.
>
> Thanks for pointing these out; we have made these changes.

---

> > ### Comment · AnonReviewer2 · 2019-11-11
> > **Thanks!**
> >
> > Thanks for the answer. The authors have addressed the main concerns I raised, and the revised version of the paper looks more convincing to me.
> >
> > If the conference had a more gradual scoring system I would have risen my score, but I am keeping it as a "weak accept", as I am hesitant to give it the maximum score. I think that the paper makes a valuable contribution, and I do not have any major concern after the revised version, which is the reason why I lean toward acceptance. However, I partly share the general feeling of reviewer 3 that the paper is not particularly exciting (limited novelty and not fully convincing results), although I am more positive about it and I still think (more strongly now) that it would make an interesting contribution to the conference.
> >
> > Two comments on the authors' rebuttal and the revised version:
> >
> > - Thanks for adding the two "rotation" baselines. However, it is surprising that none of them are able to consistently improve over the base mBERT model, which contradicts previous findings. I think that this deserves more discussion in the paper: either this approach does not work in the general case (which would be an important finding), or you are doing something differently that could explain these negative results.
> >
> > - Regarding Lample & Conneau (2019), you are right that asking to pretrain an XLM model from scratch would not be reasonable. However, you could have used the pretrained XLM models as your baseline, which would allow you to assess your method against both the MLM and the MLM+TLM variants. I think that using mBERT does not invalidate the paper, but I do see it as a weakness. Needless to say, achieving a new SOTA (by improving XLM) would have made the results much more convincing, showing that your improvements are orthogonal with those of previous work.

---

> > > ### Author Response · Authors · 2019-11-15
> > > **Thanks again for the comments.**
> > >
> > > Thanks again for the comments, which we address below:
> > >
> > > > Thanks for adding the two "rotation" baselines. However, it is surprising that none of them are able to consistently improve over the base mBERT model, which contradicts previous findings. I think that this deserves more discussion in the paper: either this approach does not work in the general case (which would be an important finding), or you are doing something differently that could explain these negative results.
> > >
> > > We are also surprised that rotation does not produce consistent improvements in XNLI. Past work using rotations to align contextual word embeddings focus on dependency parsing (Schuster et al., 2019, Wang et al., 2019), which is word-level rather than sentence-level and more syntactic than semantic compared to XNLI. Therefore, perhaps there are key differences between the tasks that makes rotation sufficient for one but not the other. We do think this discrepancy deserves further examination but are hesitant to make strong negative claims without further experimentation.
> > >
> > > > Regarding Lample & Conneau (2019), you are right that asking to pretrain an XLM model from scratch would not be reasonable. However, you could have used the pretrained XLM models as your baseline, which would allow you to assess your method against both the MLM and the MLM+TLM variants. I think that using mBERT does not invalidate the paper, but I do see it as a weakness. Needless to say, achieving a new SOTA (by improving XLM) would have made the results much more convincing, showing that your improvements are orthogonal with those of previous work.
> > >
> > > We agree and hope to explore the application of our method to XLM in future work. We think it may also be worthwhile to experiment with the recently released XLM-R (Conneau et al., 2019), which is similar to mBERT but uses much more data, more parameters, and more training time to achieve very high XNLI numbers.

---

### Official Review · AnonReviewer1 · 2019-10-26
**Official Blind Review #1**

**Rating:** 6

**Review:**

The paper proposed a pre-training method for strengthening the contextual embeddings alignment. Given parallel sentences from a different language, the authors proposed to enforce corresponding words that have a similar representation by minimizing the squared error loss. The authors also proposed to use the an regulation that prevents the learned embedding from drift too far. The authors evaluated the proposed pre-training on the contextual alignment metric and show the BERT has variable accuracy depends on the language. The proposed method improved significantly on zero-shot XNLI compares to the base model.

The paper is well written, and the proposed aligned loss makes sense and should augment the multi-lingual pre-training from a high level. The authors did a good job of analyzing the bert for multi-lingual. There some details may help the reader understand the paper better

1: Why use L2 distance as the metric function, what is the performance of using the inner product as a metric function? and what is the difference here?

2: The authors mentioned the word pairs are extracted from the existing method which may be noisy. I wonder is there any ablations study with respect to how the word pairs affect the pretraining?

3: When finetuning on zero-shot transfer, what is the finetune setting? Is there any strategy to avoid the lower layer embedding from drifting away?

4: In table 3, the Fully supervised Base Bert on English is close to the zero-shot setting and the base BERT model is better than Alignment bert, I wonder can the authors explain more on this?



**Experience Assessment:**

I have published one or two papers in this area.

**Review Assessment: Checking Correctness Of Derivations And Theory:**

I carefully checked the derivations and theory.

**Review Assessment: Checking Correctness Of Experiments:**

I carefully checked the experiments.

**Review Assessment: Thoroughness In Paper Reading:**

I read the paper at least twice and used my best judgement in assessing the paper.

---

> ### Author Response · Authors · 2019-11-10
> **Thank you for the interesting questions.**
>
> Thank you for the interesting questions, which we address below:
>
> 1: These two metric functions are quite similar because ||a - b||^2 = ||a||^2 - 2<a, b> + ||b||^2, so if the vectors remain roughly the same length, minimizing the L2 distance is similar to maximizing the inner product. To avoid blowing up the vector lengths, we would probably want to maximize the cosine similarity instead (the inner product normalized by the norm). Given that the retrieval evaluation uses a modified version of cosine similarity, optimizing this metric instead of L2 could be interesting to explore as a way to improve alignment.
>
> 2: This is a good point: it would be interesting to examine how our method fares under higher noise situations. Some possible ablations might be using expert-annotated word pairs or inserting fake word pairs to simulate noise in a controlled manner, which we have not tried yet. Given that we were more interested in precision over recall, we used the intersect method to produce less noisy word pairs, with the tradeoff of lower coverage. It’s possible that the method could benefit from a higher recall approach, where we have more word pairs but they are noisier. One reason to prefer higher precision is that we only use 250K sentences from the 2M sentences in Europarl, so we could instead just increase the number of sentences if we wanted more word pairs. But in a low-resource setting, we might have fewer parallel sentences, so a higher recall approach could make sense. Characterizing the method’s robustness to noise could help us find the optimal tradeoff between precision and recall, which we hope to explore in future work.
>
> 3: When we fine-tune on zero-shot transfer, we allow all of the weights to change, but we also use linear learning rate warmup, which might prevent some of the initial drift you are mentioning. The rest of the optimization hyperparameters are in the appendix. It’s a good point that the model could forget the alignment while fine-tuning on a downstream task, so it might be useful to maintain some sort of regularization that keeps the embeddings aligned. This approach seems worth trying and could improve accuracy.
>
> 4: Thank you for pointing this out. The varying English accuracy across the zero-shot models results from our method of model selection, where we select a model based on its average accuracy across the languages. Therefore, if a model has unusually high zero-shot accuracy early on in the training procedure, we might select that checkpoint even if it has low English accuracy. We have added a sentence in the paper to explain this point.

---

### Decision · Program_Chairs · 2019-12-19

**Decision:**

Accept (Poster)

**Comment:**

This paper proposes a method to improve alignments of a multilingual contextual embedding model (e.g., multilingual BERT) using parallel corpora as an anchor. The authors show the benefit of their approach in a zero-shot XNLI experiment and present a word retrieval analysis to better understand multilingual BERT.

All reviewers agree that this is an interesting paper with valuable contributions. The authors and reviewers have been engaged in a thorough discussion during the rebuttal period and the revised paper has addressed most of the reviewers concerns.

I think this paper would be a good addition to ICLR so I recommend accepting this paper.